# Greedy Coordinate Diffusion: Effective and Semantically Coherent Adversarial Attacks via Diffusion Guidance

Bohdan Turbal [1]   Blossom Metevier [1]   Max Springer [1]   Aleksandra Korolova [1]

## Abstract

Adversarial attacks on large language models have limited practical impact despite extensive research. Optimization-based attacks such as Greedy Coordinate Gradient (GCG) (Zou et al., 2023) produce high-perplexity, incoherent suffixes that existing defenses easily detect (Bengio et al., 2024). Moreover, attempting to enforce coherence constraints during optimization often prevents the attack from successfully eliciting the specific targeted response, resulting in low success rates against robust models. Conversely, attacks that maintain coherence often alter the semantic intent of queries; when the model complies with these altered queries, responses fail to address the adversary's original goal. In this work, we introduce Greedy Coordinate Diffusion (GCD), a novel framework that efficiently generates adversarial attacks against safety-aligned models while maintaining low perplexity and high semantic adherence to the adversary's original intent. GCD leverages the generative priors of discrete diffusion language models to guide the search for adversarial prompts that achieve semantic coherence and adherence. Unlike GCG, GCD does not require direct gradient access, allowing it to operate in a gray-box setting. We show GCD achieves highest attack success rate while remaining competitive on response-quality scores, and that the constructed adversarial prompts are detected at lower rates than other methods by perplexity-based and guard-model filters.

[1]Princeton University, Princeton, NJ, USA. Correspondence to: Bohdan Turbal <bt4811@princeton.edu>, Blossom Metevier <bmetevier@princeton.edu>, Max Springer <maxspringer@princeton.edu>, Aleksandra Korolova <korolova@princeton.edu>.

*Proceedings of the $43^{rd}$ International Conference on Machine Learning*, Seoul, South Korea. PMLR 306, 2026. Copyright 2026 by the author(s).

## 1. Introduction

The widespread deployment of large language models (LLMs) in sensitive domains has motivated work in safety mechanisms that aim to reduce the likelihood of models generating harmful outputs. On the opposing side, an active area of research is to develop adversarial prompting attacks, commonly referred to as jailbreaks, that can bypass the safety mechanisms.

Despite their effectiveness on benchmarks, practical deployment of existing jailbreak attacks remains limited. Current methods encounter three primary failure modes. A jailbreak attack incurs *high perplexity* when the adversarial prompt is unnatural and consequently is easily flagged by filters. It incurs a *jailbreak tax* when the attack succeeds only by semantically altering the original query, causing the model to comply with a benign restatement rather than the adversary's actual intent. Lastly, it can yield a low attack success rate (ASR) when it fails to reliably induce compliance at all.

These failure modes are not independent. Low ASR is often a downstream consequence of the other two: attacks that avoid high perplexity by rewriting the query tend to dilute adversarial intent, incurring a jailbreak tax; attacks that preserve intent but ignore readability produce unnatural text that modern guard models detect and block. Escaping one failure mode thus typically exacerbates another, and current methods lack the fine-grained semantic control needed to navigate this tradeoff.

Gradient-based methods such as Greedy Coordinate Gradient (GCG) (Zou et al., 2023) exemplify the perplexity failure mode, wherein they append optimized token sequences to benign queries, resulting in high ASR but unnatural suffixes that are readily detected by modern guard models. Prompt-rewriting approaches such as PAIR (Chao et al., 2023) and AutoDAN-Turbo (Liu et al., 2024) are on the opposite end, producing fluent, human-readable text but relying on conceptual rephrasing that alters the semantic intent of the query. As shown by Nikolić et al. (2025), this results in high jailbreak tax: the model complies only because the prompt has been rendered effectively benign.

Methods like COLD (Guo et al., 2024), AutoDAN (Liu et al., 2023), and Diffusion Attacker (Wang et al., 2025) attempt

to balance readability with exact token-level optimization (such as target prefix matching), but face a different failure mode wherein they treat the attack loss as a heuristic guide rather than a strict constraint and suffer from convergence issues that trade ASR for fluency. Consequently, these methods are often unable to force more protected models to generate specific target outputs, failing where more precise optimization might succeed. Our goal is therefore to combine the fluency of prompt-rewriting approaches with the fine-grained token-level control of GCG.

Human readability is essential for evading lightweight perplexity-based filters commonly deployed as a first line of defense. Token-level control is necessary to achieve high ASR through optimization. Successful attacks should preserve the adversary's original intent, so that compliance reflects the malicious query. We propose **Greedy Coordinate Diffusion (GCD)**, a gray-box optimization framework that simultaneously achieves high attack success rates, produces human-readable prompts, and preserves semantic alignment with the attacker's original intent. The key insight behind GCD is to replace gradient-based token selection with a discrete diffusion model as a proposal distribution, rather than GCG's gradient-based optimization over one-hot token embeddings which yields incoherent text. We aim to avoid the jailbreak tax via prompt-based conditioning of the diffusion model and by explicitly forcing the victim LLM to output a strictly constrained adversarial target. This limits any potential alleviation of harmfulness through target modification, ensuring the model complies with the actual malicious query rather than a semantically altered version.

By formulating prompt construction as a guided discrete diffusion process, we demonstrate that GCD constrains the optimization search space to coherent, human-readable text while still allowing for fine-grained token-level optimization. Specifically, we optimize the adversarial prompts to minimize the negative log-likelihood of a target affirmative prefix (e.g. "Sure, here is how to...:\n\n") under the victim model. The diffusion model proposes semantically coherent candidate tokens at each position, while the greedy optimization evaluates these candidates against the precise objective.

By enforcing token-level optimization to enable precise convergence, while maintaining natural language, GCD ensures that the victim model output is inherently constrained by the adversarial objective. Simultaneously, the malicious prompt is constrained by the diffusion model's generative prior, preserving the semantic intent of the original malicious query.

## 2. Related Work

**Optimization-Based Attacks.** Optimization-based attacks formulate adversarial prompt generation as an optimization problem over an attack objective, such as the likelihood of eliciting a target response prefix (Zou et al., 2023). These attacks either operate directly in the discrete token space or in an intermediate continuous space, such as over token distributions or an embedding space. GCG exemplifies the discrete approach, using gradient-based token selection to iteratively construct adversarial suffixes.

These methods achieve high ASR for simpler targets and low jailbreak tax (Nikolić et al., 2025), but face two critical limitations. First, by their greedy nature, they produce high-perplexity, unnatural text that is easily detected by perplexity-based filters (Jain et al., 2023; Hu et al., 2023). Second, each optimization step evaluates discrete candidates sampled from the projection of the gradients to the token space, which is inherently noisy, causing the optimization to fail for more intricate target objectives or more heavily defended models.

Continuous methods such as COLD (Guo et al., 2024) perform optimization over token distributions or embeddings and then project into the token space. While faster, these methods suffer from a different bottleneck: the final continuous-to-discrete projection discards precision accumulated during optimization, undermining constraint control. Furthermore, these methods cannot optimize the target cross-entropy loss effectively, treating it more as a heuristic than a precise objective and success criterion (Guo et al., 2024; 2021), limiting their ability to enforce specific response prefixes.

These limitations motivate the need for methods that constrain the search space to coherent language while preserving token-level control—the core design goal of GCD.

**Human-Readable & Diffusion-Based Attacks.** Prompt-rewriting approaches such as PAIR (Chao et al., 2023), TAP (Mehrotra et al., 2023), and GPTFuzz (Yu et al., 2023) employ attacker LLMs to iteratively refine prompts. Unlike GCG, PAIR queries the victim model for output-level compliance signals and rewrites the prompt accordingly, providing no fine-grained loss signal and limited ability to target specific response prefixes.

The deployment of guard models such as Llama Guard (Inan et al., 2023) and Qwen Guard (Zhao et al., 2025) has introduced an additional layer of defense, filtering prompts and responses before and after generation. This motivates attacks that evade the initial filtering stage while maintaining effectiveness against the victim model. GCD directly addresses this by incorporating both a perplexity penalty and a guard model loss into its composite optimization objective.

Concurrent and independent works have begun investigating diffusion models for adversarial generation.

For example, (Wang et al., 2025) employ continuous diffusion over a latent embedding space. However, continuous-space attacks have been shown to project unreliably back into discrete token space and are often unable to enforce specific response prefixes through optimization (Guo et al., 2021; 2024), limiting their optimization potential. (Lüdke et al., 2025) formulate attack generation as an amortized search problem, masking prompt tokens and conditioning on a target harmful response to recover the adversarial query via reverse diffusion. However, this approach lacks direct prompt-based control over the generation process, limiting the adversary's ability to enforce specific properties on the resulting attacks. Moreover, because it relies on the model's natural conditional sampling rather than precise greedy optimization, the method often fails on complex targets, necessitating expensive $N$-time repetitions of the entire attack process. Additionally, applying discrete diffusion models to attack standard LLMs introduces a fundamental evaluation mismatch. Because intermediate diffusion states are populated with $<|\text{mask}|>$ tokens, standard autoregressive victim models cannot accurately process them to provide optimization feedback. Although Lüdke et al. attempt to guide their sampling using target-model likelihoods, they do not propose a meaningful solution to address this token-space incompatibility.

Together, these approaches highlight two unresolved challenges: unreliable discrete projection in continuous methods, and the inability of sampling-based methods to efficiently evaluate and steer intermediate masked states. GCD addresses both by operating directly in the discrete token space, utilizing a one-shot diffusion lookahead to bridge the mask-evaluation mismatch, and applying a greedy attack-loss criterion to precisely guide the diffusion proposal at each step.

**Discrete Diffusion Models.** Diffusion Language Models have emerged as a powerful alternative to autoregressive generation, enabling non-sequential text editing by iteratively denoising corrupted token sequences (Austin et al., 2021; Ye et al., 2025). Discrete approaches have been shown to be easier to scale and to generate higher-quality language than their continuous counterparts (Austin et al., 2021; Lou et al., 2024; Sahoo et al., 2024).

Discrete diffusion models are particularly well-suited for adversarial prompts optimization for two reasons. First, unlike autoregressive models that generate text strictly left-to-right, discrete diffusion models condition on the full context (both left and right) enabling coherent token proposals at any position, thus supporting non-sequential token substitution, which is precisely the operation required for efficient ad-

versarial search (Zou et al., 2023; Jia et al., 2024). Second, they support filling in multiple positions in parallel, which enables us to quickly produce a draft during the guidance process, which would be infeasible in conventional autoregressive generation (Zou et al., 2023; Jia et al., 2024).

In standard discrete diffusion frameworks, text is generated non-autoregressively by filling in multiple tokens in parallel, typically guided by simple heuristics such as random selection or entropy-based metrics (Chang et al., 2022; Kim et al., 2025). GCD replaces these heuristics with a greedy criterion over the attack loss, explicitly steering the process toward semantically coherent, human-readable adversarial prompts that maximize the likelihood of eliciting a targeted harmful response from the victim model while simultaneously evading perplexity-based and semantic guard defenses.

We leverage this by employing the discrete diffusion model as a *proposal distribution* for adversarial prompt optimization, explicitly guiding the generation process by selecting both mask positions and replacement tokens based on a greedy criterion with respect to the attack loss.

## 3. Preliminaries

We formalize the adversarial attack setting, establish our threat model assumptions, and define the optimization objective and success criteria used throughout the paper.

**Problem Formulation.** Let $M_\theta$ be an LLM parametrized by $\theta \in \Theta \subseteq \mathbb{R}^n$, which maps a sequence of input tokens to a probability distribution over the vocabulary $\mathcal{V}$. Let $A_{adv}$ be the adversary — a mapping from a malicious intent $I_{\text{mal}}$ and a target model $M_\theta$ to an adversarial prompt $x_{\text{adv}} = (t_1, \ldots, t_L)$, where $t_i \in \mathcal{V}$ for all $i \in \{1, \ldots, L\}$, with the goal of prompting $M_\theta$ to elicit harmful behaviour aligned with $I_{\text{mal}}$. We define the harmfulness function $H : \mathcal{V}^* \times \mathcal{I}_{\text{mal}} \to [0, 1]$, which maps a model response $M_\theta(x_{\text{adv}})$ and malicious intent $I_{\text{mal}}$ to a normalized score, where 1 indicates a fully harmful response and 0 – a benign one. For notational convenience, we use $x_{\text{adv}}$ and the model response $M_\theta(x_{\text{adv}})$ interchangeably, simply writing $H(x_{\text{adv}})$ when $I_{\text{mal}}$ is clear from context.

To maximize $H(x_{\text{adv}})$ in practice, the adversary minimizes a proxy loss $\mathcal{L}_{\text{victim}}$ that is inversely related to $H(x_{\text{adv}})$. For LLM jailbreaking, it is often sufficient to enforce a specific target prefix $y_{\text{target}}$ of $M_\theta(x_{\text{adv}})$ (Zou et al., 2023); thus we define $\mathcal{L}_{\text{victim}}$ as the negative log-likelihood of $y_{\text{target}}$ conditioned on the adversarial prompt:

$$\mathcal{L}_{\text{victim}}(x_{\text{adv}}) = -\log M_\theta(y_{\text{target}} \mid x_{\text{adv}}).$$

While minimizing $\mathcal{L}_{\text{victim}}$ aims to increase $H(x_{\text{adv}})$, in practice an effective adversarial prompt must additionally remain *human-readable* and *stealthy* — that is, undetectable by de-

fense mechanisms such as perplexity-based filters (Hu et al., 2023) and trained guard models (Inan et al., 2023; Zhao et al., 2025). We therefore extend the attacker's objective to a composite loss:

$$\mathcal{L}_{\text{comp}}(x_{\text{adv}}) = \mathcal{L}_{\text{victim}}(x_{\text{adv}}) + \lambda \, \mathcal{L}_{\text{guard}}(x_{\text{adv}}) + \beta \, \mathcal{L}_{\text{PPL}}(x_{\text{adv}}),$$

$$x_{\text{adv}}^* = \underset{x_{\text{adv}} \in \mathcal{V}^L}{\operatorname{argmin}} \, \mathcal{L}_{\text{comp}}(x_{\text{adv}}),$$

where $\mathcal{L}_{\text{guard}}$ is the negative log-likelihood of $x_{\text{adv}}$ being classified as safe by a guard model $G_{\text{guard}}$ (e.g., LlamaGuard), penalizing prompts that are likely to be flagged. $\mathcal{L}_{\text{PPL}}$ is the self-perplexity of $x_{\text{adv}}$ as assigned by $M_\theta$ itself, penalizing unnatural token sequences:

$$\mathcal{L}_{\text{PPL}}(x_{\text{adv}}) = \exp\left(-\frac{1}{L}\sum_{i=1}^{L}\log p_\theta(t_i \mid t_{<i})\right),$$

where $p_\theta(t_i \mid t_{<i})$ is the probability assigned by $M_\theta$ to the $i$-th token given its preceding context (Jelinek et al., 1977). Notably, low perplexity is consistent with high-quality, human-like text (Chen & Goodman, 1999; Wenzek et al., 2020). The scalars $\lambda, \beta \geq 0$ control the relative importance of stealthiness and fluency against attack effectiveness.

**Threat Models.** Jailbreak attacks can be categorized by their assumptions regarding model access. White-box attacks assume full gradient access and use it to optimize adversarial prompts directly; while this provides the richest optimization signal, it is infeasible against proprietary or API-served models. Black-box attacks rely solely on observed text outputs, typically employing attacker LLMs for heuristic prompt rewriting (Liu et al., 2024); while broadly applicable, the sparse feedback limits fine-grained optimization. We focus on the gray-box setting: the attacker lacks access to internal weights but can leverage output token probabilities or safety scores, either exposed directly by the API or estimated via proxy models. We argue that this represents a practical step aimed at bridging the divide between sparse black-box feedback and full-information optimization. Under this threat model, we assume access to the log-probabilities (logprobs) of the victim LLM for forced or arbitrary continuations.

**Success Criteria.** Guided by the composite loss $\mathcal{L}_{\text{comp}}$, we consider a jailbreak $x_{\text{adv}}$ successful if $H(x_{\text{adv}}) > \tau_{adv}$, where $\tau_{adv}$ is a threshold determined in practice by a held-out harmfulness classifier. We further evaluate success under three defense setups:

(a) **No defense:** $x_{\text{adv}}$ is passed directly to $M_\theta$.

(b) **Perplexity filtering** (Hu et al., 2023): $x_{\text{adv}}$ is rejected if $\mathcal{L}_{\text{PPL}}(x_{\text{adv}}) > \tau_{\text{PPL}}$ for a threshold $\tau_{\text{PPL}}$ calibrated on benign human-written prompts.

---

**Algorithm 1** Greedy Coordinate Gradient (Zou et al., 2023)

1: **Input:** Initial prompt $x$, modifiable subset $\mathcal{I}$, iterations $T$, loss $\mathcal{L}$, $k$, batch size $B$
2: **for** $t = 1, \ldots, T$ **do**
3:      **for** $i \in \mathcal{I}$ **do**
4:          $\mathcal{X}_i \leftarrow \text{Top-}K(-\nabla_{e_{x_i}}\mathcal{L}(x))$
5:      **end for**
6:      **for** $b = 1, \ldots, B$ **do**
7:          $\tilde{x}^{(b)} \leftarrow x$      ▷ Initialize element of batch
8:          $\tilde{x}_i^{(b)} \leftarrow \text{Uniform}(\mathcal{X}_i)$, where $i = \text{Uniform}(\mathcal{I})$
9:      **end for**
10:     $x \leftarrow \tilde{x}^{(b^*)}$, where $b^* = \operatorname{argmin}_b \mathcal{L}(\tilde{x}^{(b)})$
11: **end for**
12: **Output:** Optimized prompt $x_{1:n}$

---

(c) **Guard model filtering**: $x_{\text{adv}}$ is rejected if $G_{\text{guard}}(x_{\text{adv}}) \neq$ `safe`, where $G_{\text{guard}}$ is Llama Guard 3-1B (Llama Team, 2024).

These three setups correspond directly to the evaluation metrics reported in Section 5.

## 4. Greedy Coordinate Diffusion

**Greedy Coordinate Gradient.** GCD is inspired by GCG (Zou et al., 2023), which approximates the discrete token search problem by using continuous gradients to identify promising token substitutions (Algorithm 1). At each iteration, GCD computes the gradient of the attack loss with respect to the one-hot embedding $e_{x_i}$ of each modifiable token, selects the top-$K$ candidates per position, and greedily updates to the candidate minimizing the loss over a batch of $B$ random substitutions: $\mathcal{X}_i = \text{Top-}K\left(-\nabla_{e_{x_i}}\mathcal{L}(x)\right), \quad x \leftarrow \tilde{x}^{(b^*)}$, where $b^* = \arg\min_b \mathcal{L}(\tilde{x}^{(b)})$.

The GCG approach has fundamental limitations that GCD is designed to address. First, the gradient is a local linear approximation of a non-linear discrete objective, introducing candidate projection error that causes optimization to fail for complex objectives or heavily defended models. Second, optimizing purely by gradient approximation produces high-perplexity, unnatural text that is detected by perplexity-based filters (Hu et al., 2023; Jain et al., 2023). Third, computing gradients requires full access to model weights, making GCG infeasible in gray-box or black-box settings.

**Greedy Coordinate Diffusion.** GCD addresses the fundamental tradeoff in adversarial attacks by replacing gradient-based token selection with a discrete diffusion model as a proposal distribution. The key insight is that discrete diffusion models, trained to reconstruct text from corrupted sequences, naturally encode the distribution of coherent language. By sampling candidate tokens from this distribution

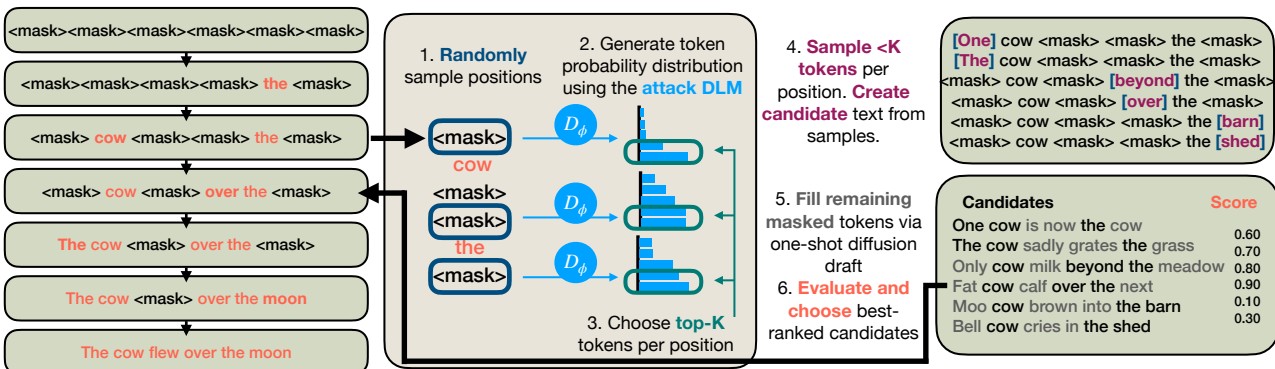

*Figure 1.* At each iteration, GCD 1) stochastically samples target positions from the current prompt; 2) queries the diffusion model $D_\phi$ for token probabilities at these indices; and 3) selects the top-$K$ candidates. Then 4) candidate sequences are formed from these tokens, and 5) any remaining masks are filled via a 1-step diffusion projection. Lastly, 6), $M_\theta$ scores the drafts to guide the next update.

rather than from gradient approximations, we constrain the optimization to explore only semantically plausible regions while maintaining token-level control. The complete procedure for GCD is detailed in Algorithm 2, where differences from GCG described in Algorithm 1 are highlighted in blue.

**High-Level Algorithm.** GCD operates via iterative token substitution on an adversarial prompt, $x_{\text{adv}}^{(i)} \in \mathcal{V}^L$. Unlike GCG which initializes with a fixed prompt, we initialize $x_{\text{adv}}^{(0)} = [<|\text{mask}|>]^L$, the sequence of fully masked tokens. This forces the diffusion model to generate semantic structure from pure noise, a task for which these models are explicitly trained (Sahoo et al., 2024). At each optimization step $t$, we perform the following:

1. **Position Sampling:** Select the positions $\mathcal{I}$ to optimize.

2. **Diffusion Proposals:** For each position, $i$, mask this position in the current $x_{\text{adv}}^{(t)}$ and query the diffusion model for the top-$K$ most probable tokens.

3. **Candidate Generation:** Create a batch of candidate sequences by randomly sampling a token from the proposal set for a sampled position.

4. **OSD (One Shot Diffusion):** conditioned on newly sampled candidate substitutions, fill remaining $<|\text{mask}|>$ tokens with unguided diffusion generation in 1 step.

5. **Loss Evaluation:** For the sampled set of candidates, compute the loss for the victim model $M_\theta$ and greedily update using the best such sample.

The remainder of this section elaborates on the key technical components.

### 4.1. Diffusion Guidance as a Proposal Distribution

**The GCD Proposal Mechanism.** A key challenge is that safety-aligned diffusion models will by default refuse to complete prompts with malicious intent. We overcome this by changing the system prompt to specify the attack direction and commands, and pre-fill response patterns with a template. The simplified form of the template looks like this: "Sure, here's your desired adversarial prompt: '$<|\text{mask}|><|\text{mask}|>...<|\text{mask}|><|\text{mask}|>$'   ", where masks are then iteratively denoised. Only the filling of this mask sub-prompt is used and fed into the target LLM during the optimization process. Full system prompt, message and template are presented in the Appendix.

To generate candidate token replacements for position $i \in \mathcal{I}$ at optimization step $t$ we query the diffusion model $D_\phi$. Specifically, we mask the token at position $i$ in the current prompt $x_{\text{adv}}^{(t-1)}$ and extract the top-$K$ most probable tokens

$$\mathcal{C}_{\text{GCD}}^{(i)} = \text{Top-K}\left(P_{D_\phi}\left(v \mid \text{mask}(x_{\text{adv}}^{(t-1)}, i)\right)[i]\right).$$

Here $\text{mask}(x, i)$ denotes the sequence $x$ with position $i$ replaced by the mask token and $P_{D_\phi}(\cdot \mid \text{mask}(x_{\text{adv}}^{(t-1)}, i))$ is the probability distribution over vocabulary tokens output by the diffusion model conditioned on the masked sequence. Unlike autoregressive models, the diffusion model conditions on the full context (both left and right), enabling coherent token proposals. Top-K selects the $K$ highest-probability tokens at position $i$.

**Early-Stage Generation.** Crucially, in early optimization steps, the context $\text{mask}(x_{\text{adv}}^{(t)}, i)$ may consist primarily of mask tokens. Unlike standard masked transformer models like BERT (Devlin et al., 2019), the diffusion model leverages its learned global prior to propose tokens that maintain long-range semantic coherence. This capability to infer structure from minimal information is essential for forming

---

**Algorithm 2** Greedy Coordinate Diffusion (Ours)

---

1: **Input:** Initial prompt $x_{1:L}$, subset $\mathcal{I}$, iterations $T$, loss weights $\lambda, \beta$, batch size $B$, Diffusion $D_\phi$, subsampling $\gamma$
2: **Definitions:** $\mathcal{X}_{\text{mask}} \subset (\mathcal{V} \cup \{\texttt{<mask>}\})^L$, $\mathcal{X}_{\text{clean}} \subset \mathcal{V}^L$
3: **Cost:** $\mathcal{L}{:}\mathcal{X}_{\text{clean}} {\to} \mathbb{R}$,
   $\qquad \mathcal{L}(\hat{x}){\leftarrow}\mathcal{L}_{\text{victim}}(\hat{x}){+}\lambda\mathcal{L}_{\text{guard}}(\hat{x}){+}\beta\mathcal{L}_{\text{PPL}}(\hat{x})$
4: **for** $t = 1 \ldots T$ **do**
5: $\quad \mathcal{I}^{(t)} {\leftarrow} \text{Sample}(\mathcal{I}, \gamma)$ ▷ Stochastic Coord. Subsample
6: $\quad$ **for** $i \in \mathcal{I}^{(t)}$ **do**
7: $\qquad \mathcal{X}_i {\leftarrow} \text{Top-}K(t)(P_{D_\phi}(\cdot \mid \text{mask}(x_{1:L}, i)))$ $\qquad$ ▷ Diffusion Proposal
8: $\quad$ **end for**
9: $\quad$ **for** $b = 1, \ldots, B$ **do**
10: $\qquad \tilde{x}_{1:L}^{(b)} \leftarrow x_{1:L}$ $\quad$ ▷ Initialize candidate with current prompt
11: $\qquad \tilde{x}_i^{(b)} \leftarrow \text{Uniform}(\mathcal{X}_i), i \in \text{Uniform}(\mathcal{I}^{(t)})$ $\qquad$ ▷ Sample candidate token
12: $\qquad \hat{x}_{1:L}^{(b)} {\leftarrow} \text{DiffusionProject}(D_\phi, \tilde{x}_{1:L}^{(b)}, \text{steps}{=}1, \tau{=}0) \in \mathcal{X}_{\text{clean}}$ ▷ 1-step diffusion projection draft on $\mathcal{X}_{\text{mask}}$
13: $\quad$ **end for**
14: $\quad b^* {\leftarrow} \text{argmin}_b \mathcal{L}(\hat{x}_{1:L}^{(b)})$
15: $\quad x_{1:L} \leftarrow \tilde{x}_{1:L}^{(b^*)}$ $\qquad$ ▷ Update token (discard projected draft)
16: **end for**
17: **Output:** Optimized prompt $x_{1:L}$

---

the attack from a fully masked initial state.

**Stochastic Coordinate Subsampling.** Standard coordinate descent evaluates potential updates for every position $i \in \{1, \ldots, L\}$ at every step. This requires $O(KL)$ victim model queries per step, which is prohibitive for long prompts. Instead we apply Coordinate Subsampling and Candidate Subsampling.

**Coordinate Subsampling**: We implement a stochastic subsampling strategy where, at each step $t$, we optimize only a random subset of coordinates. Let $\mathcal{I}^{(t)}$ be a set of active indices sampled uniformly without replacement from the full sequence indices: $\mathcal{I}^{(t)} \subset \{1, \ldots, L\}$, $\quad |\mathcal{I}^{(t)}| = \lfloor \gamma \cdot L \rfloor$, where $\gamma \in (0, 1]$ is the subsampling rate. We only generate and evaluate candidates for positions $i \in \mathcal{I}^{(t)}$. This decouples the per-step computational cost from the total prompt length, reducing complexity by a factor of $\gamma$.

**Candidate Subsampling**: Building on this coordinate-level sparsity, we apply a secondary layer of candidate-wise subsampling. From the aggregate pool of proposals across all active positions $i \in \mathcal{I}^{(t)}$, we evaluate only a proportion $\rho \in (0, 1]$ of the tokens. This directly determines the evaluation batch size $B$ used in Algorithm 2, which depends on both $\gamma$ and $\rho$ as $B = \lfloor \rho \cdot K \cdot |\mathcal{I}^{(t)}| \rfloor$, where $|\mathcal{I}^{(t)}| = \lfloor \gamma \cdot L \rfloor$.

This nested stochastic approach enables the search to maintain a wide exploration frontier while strictly limiting the number of expensive victim model queries. This further reduces the call complexity by a factor of $\rho$.

### 4.2. Bridging Token Spaces via Diffusion Look-Ahead

A fundamental challenge arises when evaluating candidate prompts: $D_\phi$ operates in a token space that includes mask tokens, while the victim model $M_\theta$ is an autoregressive LLM that cannot meaningfully process sequences containing masks. During optimization, intermediate candidate sequences may be only partially filled, containing both concrete tokens and mask positions. Directly evaluating such sequences on the victim model would produce undefined or misleading loss values and hinder attack performance, which we show by conducting ablation study with GCD and GCD-no-OSD (see Appendix).

To address this challenge, we exploit the diffusion model's ability to rapidly generate plausible completions. For each candidate sequence $x_{\text{adv}}$ that contains $k$ remaining mask tokens, we perform a one-step diffusion denoising to fill all masked positions simultaneously, conditioned on the already-committed tokens in $x_{\text{adv}}$. The result is a complete sequence with no masks, suitable for evaluation by the autoregressive victim model. This lookahead serves as a draft of what the final attack would look like if we stopped guided optimization at this point and let the diffusion model complete the sequence. This provides a coherent, semantically plausible draft completion that allows the victim model to evaluate the semantic potential of the candidate token substitution, even when the full prompt is not yet determined (see Figure 1).

**Scoring and Selection.** With this mechanism, to evaluate a candidate token $v$ at position $i$, we first construct an intermediate sequence $\tilde{x}$ by replacing the $i$-th token of $x_{\text{adv}}^{(t-1)}$ with $v$. Our scoring function for this candidate, denoted $S(v)$, becomes

$$S(v) = \mathcal{L}_{\text{comp}} \left( \text{DiffusionProject}(D_\phi, \tilde{x}) \right),$$

where the operator DiffusionProject constructs a complete candidate sequence for evaluation by performing a one-step diffusion look-ahead on $\tilde{x}$ to project the remaining masks into the discrete token space.

**Tokenizer-Agnostic Evaluation.** Once the candidate set $\mathcal{C}_{\text{GCD}}^{(i)}$ is generated, the selection of the optimal update is determined by a text-based loss evaluation. Crucially, because the proposals are converted to text strings before evaluation, GCD is *tokenizer-agnostic*. The diffusion model $D_\phi$ and the victim model $M_\theta$ need not share the same vocabulary or tokenizer for the victim to pass feedback to the diffusion generative model.

### 4.3. Final Attack Selection

We now describe a final step, Phase 2, that is omitted from Algorithm 2 pseudocode for brevity.

To maximize both attack success and response quality, GCD uses a two-phase pipeline. In Phase 1, optimization proceeds with $\mathcal{L}_{\text{comp}}$ for up to $T$ steps and min 12 steps; the attack succeeds once the victim's greedy continuation begins verbatim with $y_{\text{target}}$ (an *exact prefix match*). If Phase 1 terminates due to reaching the maximum number of iterations or a time limit without a successful match, we generate continuations for the final 8 candidate attacks from this phase.

Upon the first successful attack in Phase 1, we enter Phase 2. Here, each subsequent step that achieves a prefix match triggers a full 512-token greedy generation, which is scored by a proxy Qwen2.5-7B LLM judge on a 1–5 harmfulness scale, without directly querying the final evaluation judges (HarmBench and StrongReject). Candidates scoring $\geq 4$ — indicating substantive, real-world harmful compliance rather than hedged or off-topic content — are collected into a pool of size at least 8. To construct this pool, we evaluate the 512-token continuations of prefix-enforced attacks in batches of 16, terminating the search once at least 8 candidates qualify. If fewer than 8 adversarial attacks achieve a score of 4 or higher, we simply retain the 8 candidates with the highest scores. Finally, we use Gemini 2.5 Flash to select the final adversarial example from this pool, which is then evaluated against the final HarmBench and StrongReject judges.

**Diversity Loss.** When the victim model consistently produces evasive continuations despite the forced prefix, optimization stagnates on a narrow set of semantically similar failures. To counteract this, when the judge scores a continuation as a failure, the first token generated after $y_{\text{target}}$ accumulates a failure strike; once a token exceeds a strike threshold it is added to $S_{\text{div}}$, and the Phase 2 objective becomes:

$$\mathcal{L}_{\text{comp}}^{(2)}(x_{\text{adv}}) = \mathcal{L}_{\text{comp}}(x_{\text{adv}}) + \delta\, \mathcal{L}_{\text{div}}(x_{\text{adv}}),$$

$$\mathcal{L}_{\text{div}}(x_{\text{adv}}) = -\sum_{v \in S_{\text{div}}} \log\Big(1 - M_\theta(v \mid x_{\text{adv}},\, y_{\text{target}})\Big),$$

where $\delta \geq 0$ is the diversity coefficient and $\mathcal{L}_{\text{div}}$ is an unlikelihood term that suppresses persistently evasive first-continuation tokens, steering the search toward semantically distinct attack trajectories.

## 5. Experiments

### 5.1. Experimental Setup

**Target Models.** To ensure a comprehensive evaluation across diverse architectures and alignment strategies, we subject three distinct families of open-weight

LLMs to our attack. Following the protocols of recent work (Zou et al., 2023; Qi et al., 2025), we target: **Llama-3-8B-Instruct** (Grattafiori et al., 2024), **Qwen-2.5-7B-Instruct** (Yang et al., 2025) and **Mistral-7B-Instruct-v0.3** (Jiang et al., 2023). These models cover multiple architectures and parameter classes, with varying degrees of safety training and refusal robustness. To maintain consistency and manage computational requirements, we use 4-bit NF4 quantization for all target models throughout our experiments.

**Baselines.** Our comparison spans several threat models, evaluating GCD against white-box attacks (**GCG** (Zou et al., 2023), **GBDA** (Guo et al., 2021)), gray-box attacks (**AutoDAN** (Liu et al., 2023)), and black-box attacks (**PAIR** (Chao et al., 2023), **TAP** (Mehrotra et al., 2023), and **Inpainting** (Lüdke et al., 2025)). All baselines are executed using the hyperparameters recommended in their respective official implementations, adjusted to fit within a 1-hour compute constraint. The final adversarial attacks are selected according to the procedure defined in the original implementations. We additionally evaluate a direct request baseline (**DirectRequest**), in which the original malicious query is passed directly to the victim model without adversarial optimization or prompt transformation.

**Implementation Details.** We use the Dream-v0-Instruct-7B (Ye et al., 2025) discrete diffusion model as our generative guidance prior ($D_\phi$), keeping its weights frozen during optimization. The adversarial prompt length is set to $L = 80$ tokens. Recall from Section 4 that unlike standard GCG, which initializes with a specific string (e.g., "! ! ! !"), we initialize the prompt entirely with mask tokens ($x^{(0)} = [\texttt{<|mask|>}]^L$) to allow the diffusion model to dictate the initial semantic structure. The optimization runs for up to $T = 800$ steps (and is capped by 1 hour). We use $K = 32$ and $\delta = 0.2$. For the default and perplexity-based setups, we use $\beta = 0.1$; for the guard-model defense setup, we additionally set $\lambda = 0.2$. We use Qwen-2.5-7B as a judge for Phase 2.

To ensure efficiency for long sequences, we utilize stochastic coordinate subsampling with a rate of $\gamma = 0.25$, evaluating candidates for only 25% of positions, and further select $\rho = 0.1$ of remaining candidates for evaluation. All experiments were conducted on NVIDIA A100 (80GB) GPUs. For the perplexity-based defense, following previous literature (Jain et al., 2023) we use self-perplexity—perplexity of the prompt using victim language model.

### 5.2. Evaluation Metrics

For computational reasons, we use a stratified per-category sample of 50% of the JailbreakBench (Chao et al., 2024) dataset, further removing requests related to pedophilia due to legal concerns, resulting in 48 requests, which encompass

*Table 1.* Main Results: Attack Success Rate (ASR). We compare GCD against prominent gradient-based (GCG, GBDA) and human-readable (PAIR, TAP, AutoDAN, Inpainting) baselines across three victim models.

| VICTIM MODEL | METHOD | HB ASR (↑) | SR SCORE (↑) | HB PPL DEF | SR PPL DEF |
|---|---|---|---|---|---|
| LLAMA-3-8B | DIRECT REQUEST | 4.17% | 0.0368 | 4.17% | 0.0368 |
| | GCG | 0.00% | 0.0104 | 0.00% | 0.0000 |
| | GBDA | 0.00% | 0.0039 | 0.00% | 0.0000 |
| | PAIR | 4.17% | 0.1436 | 4.17% | 0.1436 |
| | TAP | 6.25% | 0.0553 | 6.25% | 0.0553 |
| | AUTODAN | 0.00% | 0.1568 | 0.00% | 0.1568 |
| | INPAINTING | 22.92% | 0.1575 | 22.92% | 0.1575 |
| | **GCD (OURS)** | **85.42%** | **0.7159** | **85.42%** | **0.7159** |
| QWEN-2.5-7B | DIRECT REQUEST | 4.17% | 0.1199 | 4.17% | 0.1199 |
| | GCG | 68.75% | 0.4854 | 0.00% | 0.0000 |
| | GBDA | 8.33% | 0.0934 | 0.00% | 0.0000 |
| | PAIR | 25.00% | 0.5157 | 25.00% | 0.5157 |
| | TAP | 33.33% | 0.5084 | 33.33% | 0.5084 |
| | AUTODAN | 91.67% | **0.8054** | 91.67% | **0.8054** |
| | INPAINTING | 85.42% | 0.4306 | 85.42% | 0.4306 |
| | **GCD (OURS)** | **97.92%** | 0.7789 | **97.92%** | 0.7789 |
| MISTRAL-7B | DIRECT REQUEST | 52.08% | 0.5320 | 47.92% | 0.5082 |
| | GCG | 77.08% | 0.6440 | 0.00% | 0.0000 |
| | GBDA | 54.17% | 0.5630 | 0.00% | 0.0000 |
| | PAIR | 41.67% | 0.5005 | 41.67% | 0.5005 |
| | TAP | 50.00% | 0.4909 | 50.00% | 0.4909 |
| | AUTODAN | 95.83% | 0.8140 | 95.83% | 0.8140 |
| | INPAINTING | 89.58% | 0.7381 | 89.58% | 0.7381 |
| | **GCD (OURS)** | **100.00%** | **0.8234** | **100.00%** | **0.8234** |

a wide variety of categories such as hacking, government decision-making, and fraud, among others. To strictly evaluate both effectiveness and utility, we employ the following metrics:

**HarmBench (HB) ASR:** We employ the cais/HarmBench-Llama-2-13b-cls classifier (Mazeika et al., 2024), which is specifically fine-tuned for standard and contextual behaviors. We calculate the ASR of the generated jailbreaks using binary classification and report mean ASR.

**StrongReject (SR) Score:** As demonstrated in StrongReject (Souly et al., 2024), the ASR of jailbreaks is often overestimated because some evaluators fail to measure the actual completeness of an adversarial attack's output. To address this, they fine-tune a Gemma-2 model for fine-grained harmfulness evaluation, reporting a continuous score rather than binary ASR. We use this metric as a proxy for the *jailbreak tax*.

**HB PPL DEF & SR PPL DEF:** These represent the Harm-Bench and StrongReject evaluation metrics under a perplexity defense. We filter out attacks that exceed the 85th percentile threshold of perplexity, calculated using benign prompts from the lmsys/lmsys-chat-1m dataset (Zheng et al., 2023). Attacks that do not pass the defense receive score 0 for both HB ASR and SR Score.

**HB Guard DEF & SR Guard DEF:** These represent the

HB and SR evaluation metrics under a meta-llama/Llama-Guard-3-1B (Llama Team, 2024) defense, where we consider attack to be successful only if it breaks victim LLM and remains safe under this model.

When evaluating attacks under defense, we first apply the corresponding defense filter and then compute HarmBench ASR and StrongReject score on the remaining attacks from the pool received after the Phase 2. Attacks rejected by the defense are assigned both HB ASR = 0 and SR Score = 0.

### 5.3. Experimental Results

Tables 1 and 2 present the comparative performance.

**Effectiveness & Readability.**

Across the compared target models, GCD achieves the highest HB ASR under the default attack setup. The improvement is especially prominent for Llama-3-8B, where the strongest baseline, Inpainting, achieves 22.92% ASR on HB, compared to 85.42% for GCD, and an SR score of 0.1575, compared to 0.7159 for GCD. This behavior is consistent with the other 7B-scale models, although the margin is smaller due to their initially higher vulnerability. Furthermore, GCD obtains the strongest SR performance on Llama-3-8B and Mistral-7B. On Qwen-2.5-7B, GCD achieves a competitive SR score of 0.7789, below AutoDAN's 0.8054,

*Table 2.* Defense Performance. GCD compared against GCG, GBDA, PAIR, TAP, AutoDAN, Inpainting across three victim models using Guard Defense metrics.

| Victim Model | Method | HB Guard DEF | SR Guard DEF |
|---|---|---|---|
| Llama-3-8B | Direct Request | 0.00% | 0.0000 |
| | GCG | 0.00% | 0.0000 |
| | GBDA | 0.00% | 0.0000 |
| | PAIR | 2.08% | 0.0689 |
| | TAP | 6.25% | 0.0485 |
| | AutoDAN | 0.00% | 0.0000 |
| | Inpainting | 12.50% | 0.0902 |
| | **GCD (Ours)** | **87.50%** | **0.6788** |
| Qwen-2.5-7B | Direct Request | 2.08% | 0.0058 |
| | GCG | 0.00% | 0.0000 |
| | GBDA | 0.00% | 0.0000 |
| | PAIR | 2.08% | 0.1133 |
| | TAP | 4.17% | 0.0975 |
| | AutoDAN | 0.00% | 0.0000 |
| | Inpainting | 27.08% | 0.1366 |
| | **GCD (Ours)** | **85.42%** | **0.6768** |
| Mistral-7B | Direct Request | 2.08% | 0.0085 |
| | GCG | 0.00% | 0.0000 |
| | GBDA | 0.00% | 0.0000 |
| | PAIR | 6.25% | 0.0691 |
| | TAP | 6.25% | 0.0621 |
| | AutoDAN | 2.08% | 0.0164 |
| | Inpainting | 20.83% | 0.1395 |
| | **GCD (Ours)** | **97.92%** | **0.7900** |

though outperforming AutoDAN on HB ASR (97.92% vs. 91.67%). Together, these results suggest the benefit of combining fine-grained control with goal-aligned, semantically coherent proposals. We present qualitative examples of the generated attacks in the Appendix.

Under the perplexity-based defense (HB PPL DEF and SR PPL DEF in Table 1), GCD preserves its HB ASR and SR scores across all target models. In contrast, gradient-based baselines such as GCG and GBDA largely collapse under the same filter, with their DEF values dropping to zero in most settings. Human-readable baselines such as PAIR, TAP, AutoDAN, and Inpainting are less affected by the perplexity filter, but GCD still achieves the highest HB PPL DEF on all target models.

These results suggest that diffusion-guided proposals help GCD avoid the high-perplexity artifacts typical of gradient-based attacks while retaining strong attack effectiveness.

To demonstrate the flexibility of GCD in incorporating additional optimization objectives, such as stealthiness, in Table 2 we report performance under the Llama-Guard defense, incorporating auxiliary Defense Loss objective guidance for GCD. In this setting, most baselines experience substantial degradation, with gradient-based methods again dropping to zero and Inpainting obtaining the strongest baseline per-

formance across the victim models. In contrast, GCD with defense loss guidance demonstrates robustness against the Guard defense, retaining an HB Guard DEF of 87.50% on Llama-3-8B, and achieving 85.42% on Qwen-2.5-7B and 97.92% on Mistral-7B. These results represent substantially smaller degradation than the baselines under the same defense.

### 5.4. Limitations

Our evaluation has several limitations. GCD is tested on a fixed set of open-weight victim models, and its performance may vary on proprietary models. In particular, we do not study the transferability of our attack to other models or defense mechanisms. We cap compute at 1-hour for all attack methods, which could affect results and partially provides context as to why GCG fails to converge within this timeframe, resulting in a 0.00% HB ASR on Llama-3-8B, potentially requiring more compute than other methods. Our method also assumes a gray-box setting with access to informative optimization feedback, such as token probabilities, which may not be available in black-box deployment settings. Moreover, GCD uses a fixed diffusion model as its proposal distribution, and its effectiveness might therefore depend on the quality and tokenizer behavior of that model. Our evaluation only covers perplexity- and guard-model filters. However, deployed systems may combine multiple defenses, such as automated monitoring, rate limits, and escalation to human review (OpenAI, 2025).

## 6. Conclusion

We introduced Greedy Coordinate Diffusion (GCD), a framework that replaces noisy gradient-based heuristics with generative diffusion priors to bridge the gap between the fine-grained advantages of white-box optimization and the semantic coherence of black-box strategies. Our results show that GCD can achieve highest ASR while remaining competitive on response-quality scores against aligned models and while remaining human-readable and stealthy.

This indicates that adversarial examples need not be high-perplexity statistical anomalies, and can instead lie within the high-probability token sequences that language models are optimized to produce. This suggests that defenses based primarily on surface-level statistics such as perplexity may have limited coverage on their own, and that incorporating richer notions of semantic intent could be beneficial.

## Impact Statement

This work investigates vulnerabilities in safety-aligned LLMs by proposing a method for generating adversarial prompts. The primary impact is to understand the limitations of current alignment and guardrail mechanisms, which

can inform the development of more robust defenses and evaluation protocols. The techniques studied can be misused to bypass existing safeguards, but in this work are presented in a controlled context with the intention of supporting the design of more reliably safe systems.

## Acknowledgments

We thank anonymous ICML 2026 reviewers for their constructive feedback that helped improve our work. This work was supported in part by NSF grants CNS-1956435 and CNS-2344925.

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
